# The Effect of Vitamin D Supplementation on Hepcidin, Iron Status, and Inflammation in Pregnant Women in the United Kingdom

**DOI:** 10.3390/nu11010190

**Published:** 2019-01-18

**Authors:** Vickie S. Braithwaite, Sarah R. Crozier, Stefania D’Angelo, Ann Prentice, Cyrus Cooper, Nicholas C. Harvey, Kerry S. Jones

**Affiliations:** 1MRC Elsie Widdowson Laboratory, Cambridge CB1 9NL, UK; ann.prentice@mrc-lmb.cam.ac.uk (A.P.); kerry.jones@mrc-epid.cam.ac.uk (K.S.J.); 2MRC Lifecourse Epidemiology Unit, University of Southampton, Southampton General Hospital, Southampton SO16 6YD, UK; src@mrc.soton.ac.uk (S.R.C.); sd@mrc.soton.ac.uk (S.D.); cc@mrc.soton.ac.uk (C.C.); nch@mrc.soton.ac.uk (N.C.H.); 3NIHR Southampton Biomedical Research Centre, University of Southampton and University Hospital Southampton NHS Foundation Trust, Southampton SO16 6YD, UK; 4Oxford NIHR Biomedical Research Centre, Nuffield Department of Orthopaedics, Rheumatology and Musculoskeletal Sciences, The Botnar Research Centre, University of Oxford, Oxford OX1 2JD, UK; 5NIHR BRC Nutritional Biomarker Laboratory, University of Cambridge, Cambridge CB2 0AH, UK

**Keywords:** Vitamin D, C-reactive protein, hepcidin, ferritin, inflammation, pregnancy

## Abstract

Iron and vitamin D deficiencies are common during pregnancy. Our aim was to identify whether antenatal vitamin D_3_ supplementation affects iron status (via hepcidin suppression) and/or inflammation. Using a subset of the UK multicenter Maternal Vitamin D Osteoporosis Study (MAVIDOS)—a double-blinded, randomized, placebo-controlled trial (ISRCTN82927713; EudraCT2007-001716-23)—we performed a secondary laboratory analysis. Women with blood samples from early and late pregnancy (vitamin D_3_ (1000 IU/day from ~14 weeks gestation *n* = 93; placebo *n* = 102) who gave birth in the springtime (March–May) were selected as we anticipated seeing the greatest treatment group difference in change in 25-hydroxyvitamin D (25OHD) concentration. Outcomes were hepcidin, ferritin, C-reactive protein, and α1-acid glycoprotein concentration in late pregnancy (25OHD concentration was measured previously). By late pregnancy, 25OHD concentration increased by 17 nmol/L in the vitamin D_3_ group and decreased by 11 nmol/L in the placebo group; hepcidin, ferritin, and inflammatory markers decreased but no treatment group differences were seen. In late pregnancy, positive relationships between 25OHD and hepcidin and 25OHD and ferritin in the placebo group were observed but not in the treatment group (group × 25OHD interaction, *p* < 0.02). Vitamin D_3_ supplementation had no effect on hepcidin, ferritin, or inflammatory status suggesting no adjunctive value of vitamin D_3_ in reducing rates of antenatal iron deficiency.

## 1. Introduction

Vitamin D deficiency and anemia frequently coexist [1,2,3,4,5]. A number of mechanisms could explain this association [1] and recent data provide evidence for a direct role of vitamin D in the suppression of hepcidin: the primary regulator of systemic iron homeostasis. Hepcidin inhibits and ultimately degrades ferroportin, the transmembrane protein that transports iron, and therefore controls the amount of iron absorbed in the intestine and released from cellular storage. Hepcidin is suppressed when iron status is low to maximize dietary iron absorption and the release of iron from stores. Conversely, inflammation causes an increase in hepcidin concentration that reduces dietary iron absorption and the capacity for iron egress from cells, resulting in decreased hemoglobin concentrations [6]. 

In vitro work has shown a 1,25-dihydroxyvitamin D (1,25(OH)_2_D; the active metabolite of vitamin D)-induced, dose-dependent decrease in hepcidin expression [7,8]. The identification of the vitamin D response element (VDRE) on the human hepcidin promoter also supports the direct effect of 1,25(OH)_2_D [8]. Recent in vivo studies in healthy adults further support an effect of both single and longer-term high doses of vitamin D on hepcidin regulation. An increase in 25-hydroxyvitamin D concentration (25OHD; the status marker of vitamin D) of 68 to 109 nmol/L 72 h after an oral dose of 100,000 IU vitamin D_2_ was accompanied by a 34% decrease in hepcidin concentration (*n* = 7) [8]. A further study found a 73% decrease in hepcidin concentration one week after a single dose of 250,000 IU vitamin D_3_ (*n* = 18) [9]. In early-stage, chronic kidney disease patients, the percentage increase in 25OHD concentration after 3 months of vitamin D_3_ supplementation (50,000 IU vitamin D_3_/week) was associated with an inverse change in hepcidin concentration (*n* = 38) [7]. In addition to the effects on hepcidin concentration, vitamin D may also have immunomodulatory effects which may in turn alter hepcidin expression and iron status [10,11,12]. 

Pregnant women are at high risk of anemia [13,14] due to the high iron demands of pregnancy, particularly in low-resource settings, and data suggest associations between low hemoglobin, low iron status, and adverse birth outcomes [15]. Improvements in iron stores and hemoglobin concentrations are associated with increases in birth weight [16]. There is conflicting evidence as to whether C-reactive protein (CRP), as a marker of inflammation, rises [17] or remains the same during pregnancy [18]. However, inflammation may be relevant in pregnant women affected by conditions such as obesity or pre-eclampsia, known to be associated with inflammatory states [19]. Low vitamin D status may be a contributing factor to iron deficiency anemia via direct effects on hepcidin or through potential inflammatory effects of low vitamin D status. 

The effect of vitamin D supplementation on iron status has not been investigated in pregnant women. Through the mechanisms outlined above, adequate vitamin D status may be necessary for optimal hepcidin function and may help to provide protection against iron deficiency, potentially offering a complementary approach to combat anemia during pregnancy. 

The aim of this current study was to examine if daily vitamin D_3_ supplementation (cholecalciferol, 1000 IU/day) in pregnancy suppresses hepcidin concentration and/or affects iron body stores (ferritin) and inflammation (CRP and α1-acid glycoprotein (AGP)) compared to placebo.

## 2. Materials and Methods 

### 2.1. Study Design and Participants

The original MAVIDOS study (Maternal Vitamin D Osteoporosis Study) has been described previously in full [20,21]. In brief, MAVIDOS was a multicenter, double-blinded, randomized, placebo-controlled trial conducted between October 2008 and February 2014 and is registered in the International Standard Randomized Control Trial Registry (ISRCTN 82927713) and the European Clinical Trials Database (EudraCT 2007-001716-23). Women over the age of 18 years with a singleton pregnancy were recruited at their early pregnancy ultrasound scan. Women were screened for vitamin D status, and only those with 25OHD concentrations between 25 and 100 nmol/L and who were not currently taking dietary supplements with more than 400 IU of vitamin D/day were eligible. Exclusion criterion included any known metabolic or chronic disease known to interfere with bone metabolism, taking medication that might interfere with intrauterine growth, foetal anomalies at the 18–21 weeks ultrasound scan, and diagnosis of cancer in the last 10 years. Women were screened for eligibility at 10–17 weeks gestation and were then randomized to receive oral vitamin D_3_ (cholecalciferol 1000 IU/day; *n* = 565) or matched placebo (*n* = 569) (Merck KGaA, Darmstadt, Germany) from ~14 weeks gestation until delivery. Randomization was achieved by a computer-generated sequence in randomly permuted blocks of ten, and both the participants and research team were masked to treatment allocation. 

Vitamin D status (as measured by circulating 25OHD concentration) in the UK population changes markedly depending on season; reflecting the availability of UVB irradiation and capacity for endogenous vitamin D synthesis. During the months of September to March/April [22] there is little capacity for endogenous synthesis and so maintenance of vitamin D status relies on dietary or supplemental sources of vitamin D. Therefore, women from the MAVIDOS study who gave birth in the springtime (March–May) and with a blood sample available at both early (~15 weeks) and late (~34 weeks gestation) pregnancy (*n* = 195) were selected for this substudy as we anticipated seeing the greatest treatment group difference in change in 25OHD concentration from early to late pregnancy and minimal endogenous vitamin D production. 

Ethical approval for the original trial was obtained by the Southampton and South West Hampshire Research Ethics Committee and written informed consent was obtained from all participants. The additional analyses measured as part of this sub-study were within the remit of the original ethics application. 

### 2.2. Anthropometry and Characteristics

Maternal age, self-reported ethnicity, parity, and gestational age (determined by ultrasound) were recorded in early pregnancy and weight and height were recorded in early and late pregnancy. Supplement use within the last 3 months was recorded using a questionnaire in early and late pregnancy and was coded into whether or not the supplement included iron. As determined by the return of empty packs, 87% of women took at least 80% of the treatment tablets provided. 

### 2.3. Biochemical Analysis

Venous blood samples were collected in early and late pregnancy into lithium heparin-coated tubes and the plasma were stored at −80 °C for subsequent analysis at MRC Elsie Widdowson Laboratory, Cambridge, UK. Hepcidin was measured by ELISA (Hepcidin 25 bioactive HS, DRG Diagnostics, Marburg, Germany; inter-assay % coefficient of variation (%CV) between 8–13) and ferritin, CRP, and AGP was measured on the automated platform by Dimension Xpand, Siemens (ferritin and high sensitivity-CRP kits from Siemens Healthcare, Erlangen, Germany, and AGP kits from Sentinel Diagnostics, Milano, Italy. Intra-assay %CV was <6 for ferritin and CRP and <11 for AGP). 25OHD had been measured as part of the main trial by radioimmunossay (Liaison, RIA automated platform, Diasorin, Stillwater, MN, USA) as previously described and was DEQAS accredited [21]. 

### 2.4. Statistical Analysis 

Statistical analysis was performed using Stata 14.1 (Stata Statistics and Data Analysis, College Station, TX, USA) on an intention to treat basis. Variables were assessed for normality. Normally distributed data are presented as mean (standard deviation (SD)) and non-normally distributed data are presented as median (interquartile range (IQR)). Treatment group differences were assessed using 2-sample *t*-tests for normally distributed variables, Mann–Whitney U tests for non-normally distributed variables and chi-squared tests for categorical variables. Proportions of iron deficient women were compared between treatment groups using chi-squared tests. Changes in variables over time were tested within each treatment group using paired *t*-tests for normally distributed, Wilcoxon signed-rank tests for non-normally distributed and McNemar’s tests for categorical variables. Differences between treatment groups in changes in characteristics were tested for using regression models with late pregnancy value as the outcome and early pregnancy value and treatment group as predictors, to account for regression to the mean. Linear regression models were fitted with 25OHD concentration as the predictor and hepcidin, ferritin, CRP, and AGP concentrations as outcomes. Hepcidin, ferritin, and CRP were transformed to normality using Fisher–Yates transformation [23] and are presented in SD units. The presented coefficients are those associated with a 10 nmol/L change in 25OHD concentration. Percentage changes in hepcidin and 25OHD concentrations from early to late pregnancy were calculated and transformed to normality using the Fisher–Yates transformation. Percentage change in 25OHD concentration was examined as a predictor of percentage change in hepcidin concentration using a linear regression model. CRP > 5 mg/L [24] and/or AGP > 500 mg/L were considered indicative of inflammation. There were no AGP concentrations >1000 mg/L so the lower cut-off of 500 mg/L was used [24]. Ferritin concentrations <15 μg/L were considered indicative of iron deficiency [25]. Thresholds for 25OHD of <25 nmol/L and <50 nmol/L were used to assess vitamin D status [26]. 

## 3. Results

### 3.1. Early Pregnancy (Pre-Supplementation)

Women giving birth in spring were mean (SD) 30 (5) years old, predominantly white, 47% nulliparous, and were seen at a mean (SD) of 15.7 (1.0) weeks gestation (Table 1). Ninety-two per cent of women reported the use of supplements over the previous three months and 60% of these women reported taking supplements that contained iron. Ten per cent of women were iron deficient (ferritin < 15 µg/L), 12% had 25OHD concentrations <25 nmol/L, and 66% <50 nmol/L and there were no treatment group differences. The inclusion of women with 25OHD < 25 nmol/L is explained by the time between the screening blood sample and early pregnancy blood sample. Of women who had 25OHD <25 nmol/L, 17% were also iron deficient compared to 9% in women with 25OHD >25 nmol/L (*p* = 0.19). Similarly, using a 25OHD concentration < or > than 50 nmol/L, 11% and 8% of women respectively also were iron deficient (*p* = 0.44). 

### 3.2. Late Pregnancy 

Women were seen at a mean (SD) of 34.7 (0.8) weeks gestation in late pregnancy (Table 2). Reported supplement use decreased in both treatment groups to 62% in total. Of those taking supplements, 98% in the vitamin D_3_ group contained iron, compared to 89% in the placebo group (*p* = 0.03). Hepcidin, ferritin, CRP, and AGP concentrations decreased in both groups by late pregnancy (Table 2 and Table 3). By late pregnancy, 67% of women were iron deficient (*p* = 0.44). Mean 25OHD concentration increased by 17 nmol/L in the vitamin D_3_ group and decreased by 10 nmol/L in the placebo group (Table 3). Correspondingly, the percentage of women with 25OHD concentration <25 and <50 nmol/L increased to 50 and 83% (*p* < 0.001), respectively, in the placebo group (Table 2), and decreased in the vitamin D_3_ group to 11 and 24% (*p* < 0.0001), respectively. Of the 61 women who had 25OHD <25 nmol/L, 79% were also iron deficient compared to 61% of the 132 women with 25OHD >25 nmol/L (*p* = 0.02). Using a 25OHD concentration < or > than 50 nmol/L, 76% and 56% of women respectively were iron deficient (*p* = 0.004). 

### 3.3. Regression Analysis

In early pregnancy, 25OHD concentration was not significantly associated with hepcidin or ferritin concentrations and there was no difference in these relationships by treatment group (Table 4). In late pregnancy, 25OHD concentration was significantly positively associated with hepcidin concentration. However, there was a significant interaction between treatment group and 25OHD concentration (*p* = 0.02), such that there was a positive association between 25OHD concentration and hepcidin in the placebo group (Beta Coefficient (95% Confidence Interval) per 10 nmol/L change in 25OHD): 0.14 (0.04 to 0.24) hepcidin SDs, *p* = 0.006) but no association in the vitamin D_3_ group (−0.02 (−0.11 to 0.07), *p* = 0.63) (Figure 1 & Table 4). A similar interaction (*p* = 0.003) was found between late pregnancy 25OHD concentration and ferritin in the placebo group (0.25 (0.17, 0.33) ferritin SDs, *p* < 0.001) but not in the vitamin D_3_ group (0.06 (−0.03, 0.15), *p* = 0.19) (Figure 1 & Table 4). There was no significant treatment group interaction between 25OHD and CRP or AGP in early or late pregnancy (Figure 1 & Table 4). When the groups were analyzed together, there was no significant relationship between percentage change in hepcidin and percentage change in 25OHD (beta (95% CI) SD per SD = 0.06 (−0.09 to 0.20), *p* = 0.44) and no significant treatment group by 25OHD concentration interaction (*p* = 0.32). 

## 4. Discussion

Antenatal supplementation with vitamin D_3_ (cholecalciferol, 1000 IU/day) in women due to give birth between March and May in the UK, increased 25OHD concentrations in the supplement group from early to late pregnancy by 17 nmol/L while the placebo group experienced a significant seasonal decrease in 25OHD concentrations (−11 nmol/L). Despite a 40 nmol/L difference between the two groups by late pregnancy, no group differences were seen in hepcidin, ferritin, or markers of inflammation.

Rates of iron deficiency (ferritin <15 µg/L) increased significantly in both groups to ~70% by late pregnancy, while hepcidin, CRP, and AGP concentrations decreased significantly. This decrease in ferritin is highly likely to indicate a corresponding decrease in hemoglobin concentrations and an increase in anemia prevalence [17,27], although this was not verified due to a lack of stored whole blood samples. In addition, the decrease in hepcidin is likely to reflect the increased requirement for iron mobilization from stores and iron intestinal absorption as iron demands increase across pregnancy. 

As with other studies, this study highlights the finding that iron deficiency is often associated with lower vitamin D concentration [2,3,5]. This was seen in late pregnancy where 79% of women with 25OHD <25 nmol/L were also iron deficient compared to 61% of women with iron deficiency and 25OHD >25 nmol/L. 

Smith et al. randomized 28 healthy individuals in Atlanta, Georgia, USA, to receive one oral bolus of vitamin D_3_ (250,000 IU) or placebo [9]. Similar to our study, 75% of participants had 25OHD concentrations <50 nmol/L (20 ng/mL) at baseline and the bolus dose increased 25OHD concentrations by ~21 nmol/L in the treatment group at the 1 week follow-up. Unlike our study, however, Smith et al. found that those in the treatment group had a ~73% decrease in hepcidin concentration with no significant change detected in the placebo group and no change in ferritin or inflammatory markers in either group, 1 week post treatment. *HAMP*, the gene encoding hepcidin, is known to contain the vitamin D response element in its promoter region [8]. Smith et al. speculated that the vitamin D bolus dose resulted in a direct suppression of *HAMP* gene expression and therefore a reduction in hepcidin concentration rather than indirectly acting through inflammation [9]. 

The major differences in study design between the study by Smith et al. and the current study were (1) population (non-pregnant versus pregnant), (2) mode of vitamin D supplementation (250,000 IU bolus versus 1000 IU/day), (3) follow-up time (1 week versus 19 weeks), and (4) Smith et al. observed no changes in 25OHD concentration, markers of iron status, or markers of inflammation in the placebo group over time. The metabolic demands of pregnancy and pregnancy-related changes in vitamin D and iron metabolism (and their binding proteins) [28,29] may influence vitamin D–hepcidin–iron interactions. In addition, the kinetics and metabolism of bolus versus frequent, lower doses of vitamin D are different [30]. Therefore, while the overall change in 25OHD concentration in the treatment arm of the two studies was similar at follow-up, it is possible that the effect of vitamin D supplementation on hepcidin depends on the dose and/or frequency of vitamin D given. 

No group difference was found in hepcidin or ferritin concentrations in late pregnancy or in change in hepcidin or ferritin between early and late pregnancy. However, there was a significant group interaction between 25OHD concentration and hepcidin and between 25OHD and ferritin in late pregnancy. Interestingly, in late pregnancy, the positive relationships between 25OHD and hepcidin and between 25OHD and ferritin that were seen in the placebo group were not observed in the vitamin D_3_ group suggesting an effect of supplementation on this relationship. Therefore, there may be a relationship between 25OHD concentration and iron metabolism when 25OHD concentrations are lowest and not at higher concentrations. This finding differs from that of Thomas et al. who reported no relationship between 25OHD and hepcidin concentrations in non-supplemented adolescent pregnant women from the USA at ~26 weeks gestation and at delivery [1]. However, whilst vitamin D status in early pregnancy and in the supplemented group in late pregnancy were similar to the Thomas study, late pregnancy vitamin D status in the placebo group was notably lower, and further supports a concentration-dependent relationship between vitamin D and iron metabolism. Other non-hepcidin mediated mechanisms may explain associations observed between vitamin D and iron status. Thomas et al. found that the relationship between 25(OH)D and hemoglobin concentration was mediated by erythropoietin (EPO), with an inverse relationship between 25(OH)D and EPO in mid-gestation and at delivery [1]. 

The current study found that CRP decreased as pregnancy progressed, in contrast to some studies that have suggested an increase [17] or no change in CRP throughout pregnancy [18]. The impact of inflammation on vitamin D status [26] and the potential of vitamin D to mediate the immune response and temper inflammation is of interest [10,11]. However, we found no effect of vitamin D supplementation on CRP or AGP, suggesting that vitamin D_3_ supplementation did not impact inflammatory pathways. 

There were a number of limitations to this analysis. Firstly, whilst ferritin concentration is the standard World Health Organization measure of iron deficiency [24,25], its interpretation, in common with other markers of iron status, can be affected by inflammation and pregnancy-related hemodilution. Whilst there is uncertainty over ferritin cut-offs for iron deficiency in pregnancy, as a marker of change in iron status, ferritin remains a sensitive biomarker for body iron stores [31,32]. Hepcidin has been suggested an alternative marker of iron status in pregnant women [17] but additional confirmatory research is required. Secondly, other biomarkers of iron status, erythropoiesis, inflammation or vitamin D (i.e., 1,25(OH)_2_D) were not measured and so alternative mechanisms were not explored. Thirdly, increases in plasma volume as well as metabolic and hormonal changes during pregnancy may confound interpretation of biomarkers of nutritional status. However, the placebo controlled nature of the study should have accounted for most of these changes. Fourthly, we observed a difference between groups in the proportion of women who took iron-containing supplements in late pregnancy which was higher in the vitamin D_3_ group. However, the difference was small (*n* = 3) and the method used to record supplement use was simple; whilst the brand of the supplement was recorded and coded for the presence or absence of iron in its formulation, the frequency and length or supplement use was not recorded and therefore it was not possible to determine the amount of iron in the supplement nor the amount of supplement consumed. Finally, 94% of women in this substudy were of white, self-reported ethnicity and the participants were screened for low and high 25OHD concentration before inclusion in the trial. It is possible that a larger effect on hepcidin would be detected at lower 25OHD concentrations. 

## 5. Conclusions

This study is the first analysis of data from an RCT investigating the effect of vitamin D supplementation on iron and inflammation in pregnancy. The study indicated that 1000 IU/day of vitamin D_3_ (cholecalciferol) in pregnancy had no effect on hepcidin or other markers of iron status and inflammation compared to placebo and suggests that 1000 IU/day of vitamin D would not be beneficial in ameliorating iron deficiency in pregnancy. 

## Figures and Tables

**Figure 1 nutrients-11-00190-f001:**
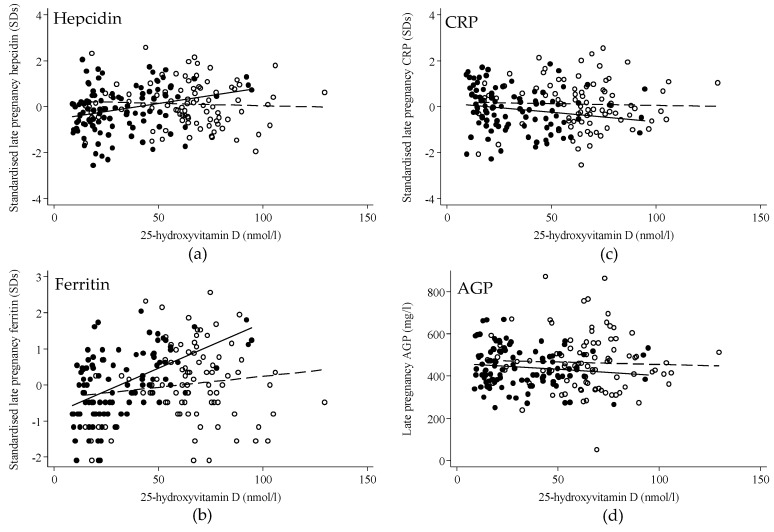
Scatterplot of 25-hydroxyvitamin D against markers of iron and inflammation in late pregnancy by treatment group. Scatterplots of 25OHD concentration in late pregnancy with hepcidin (**a**), ferritin (**b**), CRP (**c**), and AGP (**d**). Vitamin D_3_-supplemented group: open circles, dashed line; placebo group: filled circles, solid line. Regression analysis found a significant 25-hydroxyvitamin D × group interaction for hepcidin (*p* = 0.02) and ferritin (*p* = 0.003) but not for CRP (*p* = 0.32) and AGP (*p* = 0.66). For hepcidin (SDs) the placebo group beta coefficient (95% CI) was 0.14 (0.04 to 0.24) nmol/L, *p* = 0.006 and −0.02 (−0.11 to 0.07) nmol/L, *p* = 0.63 in the vitamin D_3_ group. For ferritin (SDs) the placebo group beta coefficient (95%) was 0.25 (0.17 to 0.33) nmol/L, *p* < 0.0001) and 0.06 (−0.03 to 0.15), *p* = 0.19 for the vitamin D_3_ group.

**Table 1 nutrients-11-00190-t001:** Maternal early pregnancy outcomes ^1^.

Measures in Early Pregnancy	All	Placebo	Vitamin D_3_
	*n* = (195)	*n* = (102)	*n* = (93)
White ethnicity, *n* (%)	182 (94%)	95 (93%)	87 (95%)
Previous children, *median* (*IQR*)	1 (0, 1)	1 (0, 1)	1 (0, 1)
Nulliparous, *n* (%)	91 (47%)	49 (49%)	42 (46%)
Age (years), *mean* (*SD*)	30.4 (5.2)	30.8 (5.2)	29.9 (5.2)
Gestation (weeks), *mean* (*SD*)	15.7 (1.0)	15.8 (1.0)	15.6 (1.0)
Weight (kg), *median* (*IQR*)	69.6 (61.5, 80.0)	70.5 (62.4, 79.6)	68.0 (60.7, 81.0)
Height (m), *mean* (*SD*)	1.66 (0.07)	1.66 (0.07)	1.66 (0.07)
BMI (kg/m^2^), *median* (*IQR*)	24.7 (22.1, 28.7)	24.9 (22.0, 28.8)	24.6 (22.1, 28.6)
Supplement use, *n* (%)	178 (92%)	92 (90%)	86 (93%)
Iron supplement use, *n* (% *of those using supplements*)	106 (60%)	49 (53%)	57 (66%)
CRP (mg/L), *median* (*IQR*)	5.4 (3.1, 8.3)	4.2 (2.9, 7.8)	6.2 (3.6, 11.5)
CRP > 5 mg/L, *n* (%)	103 (53%)	48 (47%)	55 (60%)
AGP (mg/L), *mean* (*SD*)	506 (115)	500 (109)	512 (122)
AGP > 500 mg/L, *n* (%)	96 (49%)	50 (49%)	46 (49%)
Hepcidin (μg/L), *median* (*IQR*)	7.3 (3.0, 16.7)	7.4 (2.5, 16.5)	6.9 (3.3, 16.8)
Ferritin (μg/L), *median* (*IQR*)	38 (25, 60)	43 (26, 62)	35 (25, 55)
Ferritin < 15 μg/L, *n* (%)	19 (9.7%)	11 (10.8%)	8 (8.6%)
25-hydroxyvitamin D (nmol/L), *mean* (*SD*)	44.1 (16.0)	42.5 (15.8)	45.7 (16.2)
25-hydroxyvitamin D < 25 nmol/L, *n* (%)	23 (12%)	15 (15%)	8 (9%)
25-hydroxyvitamin D < 50 nmol/L, *n* (%)	128 (66%)	70 (69%)	58 (62%)

^1^ Normally distributed variables are presented as mean Standard Deviation (SD), non-normally distributed variables as median (IQR), and categorical variables as *n* (%). There were no significant differences between groups (*p* > 0.05). Abbreviations: BMI, body mass index; CRP, C-reactive protein; AGP, α1-acid glycoprotein.

**Table 2 nutrients-11-00190-t002:** Maternal late pregnancy outcomes ^1^.

Measures in Late Pregnancy	All	Placebo	Vitamin D_3_	*p*-Value
	*n* = (195)	*n* = (102)	*n* = (93)	
Age (years), *mean* (*SD*)	30.6 (5.3)	31.2 (5.2) ^d^	30.0 (5.3) ^d^	0.13
Gestation (weeks), *mean* (*SD*)	34.7 (0.8)	34.6 (0.6) ^d^	34.8 (0.9) ^d^	0.06
Weight (kg), *median* (*IQR*)	78.8 (71.1, 90.4)	78.5 (73.2, 90.0) ^d^	79.2 (69.9, 92.8) ^d^	0.69
BMI (kg/m^2^), *median* (*IQR*)	28.4 (25.5, 32.9)	28.6 (25.5, 32.5) ^d^	28.3 (25.4, 32.9) ^d^	0.71
Supplement use, *n* (%)	119 (62%)	61 (61%) ^d^	58 (63%) ^d^	0.77
Iron supplement use, *n* (% *of those using supplements*)	111 (93%)	54 (89%) ^a^	57 (98%) ^b^	0.03
CRP (mg/L), *median* (*IQR*)	3.9 (2.5, 7.3)	3.8 (2.3, 6.4) ^c^	4.1 (2.6, 8.1) ^c^	0.16
CRP > 5 mg/L, *n* (%)	69 (38%)	31 (32%) ^b^	38 (43%) ^b^	0.13
AGP (mg/L), *mean* (*SD*)	452 (115)	442 (93) ^d^	463 (135) ^b^	0.20
AGP > 500 mg/L, *n* (%)	57 (29%)	28 (27%) ^c^	29 (32%) ^b^	0.53
Hepcidin (μg/L), *median* (*IQR*)	0.97 (0.79, 1.99)	0.93 (0.74, 1.57) ^d^	0.99 (0.84, 2.30) ^d^	0.19
Ferritin (μg/L), *median (IQR*)	10 (7, 18)	10 (7, 17) ^d^	10 (8, 18) ^d^	0.94
Ferritin < 15 μg/L, *n* (%)	129 (67%)	65 (64%) ^d^	64 (70%) ^d^	0.44
25-hydroxyvitamin D (nmol/L), *median* (*IQR*)	47.1 (21.4, 66.8)	25.2 (16.9, 45.8) ^d^	64.6 (52.0, 75.7) ^d^	0.0001
25-hydroxyvitamin D < 25 nmol/L, *n* (%)	61 (31%)	51 (50%) ^d^	10 (11%)	0.001
25-hydroxyvitamin D < 50 nmol/L, *n* (%)	107 (55%)	85 (83%) ^b^	22 (24%) ^d^	0.001

^1^ Normally distributed variables are presented as mean (SD), non-normally distributed as median (IQR) and categorical variables as *n* (%). *p*-value obtained from 2-sample *t*-test for normally distributed variables, Mann–Whitney U tests for non-normally distributed variables and chi-squared tests for categorical variables to test for differences between groups at late pregnancy. ^a,b,c,d^ indicates that late pregnancy variable is significantly different than early pregnancy variable by *p* < 0.05, 0.01, 0.001, and 0.0001, respectively. This is calculated from paired *t*-tests for normally distributed, Wilcoxon signed-rank tests for non-normally distributed, and McNemar’s tests for categorical variables.

**Table 3 nutrients-11-00190-t003:** Maternal change in characteristics over pregnancy (late–early pregnancy) ^1^.

Change in Characteristic	All	Placebo	Vitamin D_3_	*p*-Value
	*n* = (195)	*n* = (102)	*n* = (93)	
Maternal pregnancy weight (kg), *mean* (*SD*)	9.8 (3.6)	9.7 (3.5)	9.8 (3.8)	0.65
Maternal pregnancy BMI (kg/m^2^), *mean* (*SD*)	3.6 (1.3)	3.6 (1.2)	3.5 (1.4)	0.44
CRP (mg/L), *mean* (*SD*)	−1.6 (10.3)	−1.8 (9.4)	−1.4 (11.2)	0.12
AGP (mg/L), *mean* (*SD*)	−54 (126)	−58 (101)	−50 (149)	0.76
Hepcidin (μg/L), *median* (*IQR*)	−5.1 (−14.1, −1.3)	−5.6 (−14.9, −1.1)	−4.3 (−11.0, −1.6)	0.10
Ferritin (μg/L), *mean* (*SD*)	−31.9 (58.5)	−36.5 (41.2)	−26.9 (72.9)	0.67
25-hydroxyvitamin D (nmol/L), *mean* (*SD*)	2.6 (27.1)	−10.7 (17.9)	17.1 (28.0)	<0.0001

^1^*p*-value obtained from regression models with late pregnancy value as the outcome and early pregnancy value and treatment group as predictors, to account for regression to the mean.

**Table 4 nutrients-11-00190-t004:** Early pregnancy 25-hydroxyvitamin D (per 10 nmol/L) as a predictor of concentrations in early and late pregnancy in all women ^1^.

All Women (*n* = 195)	Beta (95% CI)	*p*-Value	*n*	Interaction *p*-Value *
Early pregnancy: hepcidin (SDs)	0.02 (−0.07, 0.11)	0.67	195	0.99
Late pregnancy: hepcidin (SDs)	0.06 (0.01, 0.11)	0.03	195	0.02
Early pregnancy: ferritin (SDs)	0.06 (−0.02, 0.15)	0.16	195	0.85
Late pregnancy: ferritin (SDs)	0.09 (0.04, 0.15)	<0.001	193	0.003
Early pregnancy: CRP (SDs)	−0.10 (−0.19, −0.02)	0.02	193	0.23
Late pregnancy: CRP (SDs)	−0.01 (−0.06, 0.05)	0.85	184	0.32
Early pregnancy: AGP (mg/L)	−5.3 (−15.4, 4.9)	0.31	195	0.06
Late pregnancy: AGP (mg/L)	−0.1 (6.4, 6.2)	0.97	194	0.66

^1^ Regression models with 25-hydroxyvitamin D concentrations (per 10 nmol/L) as the exposure and biochemical measures as the outcomes (in 8 separate models). * *p*-value for interaction between 25-hydroxyvitamin D concentration and treatment group; *p* < 0.05 indicates a significant difference in slope between the two groups. Hepcidin, ferritin, and CRP concentrations were transformed to normality using Fisher–Yates transformation and are presented in SD units.

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
