# Peer review of "The Effect of Vitamin D Supplementation on Hepcidin, Iron Status, and Inflammation in Pregnant Women in the United Kingdom"

_nutrients, 2019, doi:10.3390/nu11010190_

Reviewer 1 Report

An interesting piece of research, conducted well in terms of methodology. I do not raise any objections.

Minor inaccuracies regarding terminology ought to be corrected in the manuscript:

- the authors use the term “vitamin D3 group” in the text once (e.g. line 168, 171, 174, etc.), and the “vitamin D group” then (line 32). Moreover, the spelling of  vitamin D differs; once it is  “D3” (line 23, 168, 171, etc.), and “D3”  spelled with  the subscript another time (line 264, 280, 294). I propose adopting one term in the whole text of the article.

The abbreviation AGP in line 77 ought to be explained, as it is not explained anywhere earlier. The explanation of the abbreviation appears as late as in  line 120. 

A minor correction ought to be made concerning annotating results. The authors say that “25OHD decreased by 10 nmol/L” (line 32, 171, 195), and “by 11 nmol/L” (line 226)   another time.    

Author Response

Thank you for these suggestions. We have made changes accordingly.

Reviewer 2 Report

Major comment:

Given that the authors state that “The aim of this current study was to examine if daily vitamin D3 supplementation (cholecalciferol, 1000 IU/day) in pregnancy suppresses hepcidin concentration and/or affects iron body stores (ferritin) and inflammation (CRP and AGP) compared to placebo”, how do they explain that the placebo group during winter with lower 25OHD had a greater decline in Hepcidin that approximated significance at p=0.10 (Table 3) and the positive correlation between 25OHD and hepcidin in late pregnancy (as shown by the positive beta coefficient Table 4; being more pronounced in the placebo group in Figure 1, as reflected by the interaction effect in the regression analysis)?

Minor comment:

Lines 180-182: One assumes that this sentence refers to the P-values in the right column being a comparison of the placebo-group and the VitD3 group – but it is not stated.

Line 282: Another limitation that the authors may wish to include is the non-measurement of 1,25-dihydroxyvitamin D, especially given the mention of this hormonal form of vitamin D in Lines 49-52.

Author Response

Major comment:

Given that the authors state that “The aim of this current study was to examine if daily vitamin D3 supplementation (cholecalciferol, 1000 IU/day) in pregnancy suppresses hepcidin concentration and/or affects iron body stores (ferritin) and inflammation (CRP and AGP) compared to placebo”, how do they explain that the placebo group during winter with lower 25OHD had a greater decline in Hepcidin that approximated significance at p=0.10 (Table 3) and the positive correlation between 25OHD and hepcidin in late pregnancy (as shown by the positive beta coefficient Table 4; being more pronounced in the placebo group in Figure 1, as reflected by the interaction effect in the regression analysis)?

Thank you for your comment. This was an unexpected finding. We hypothesis that perhaps vitamin D influences hepcidin and iron metabolism when 25OHD concentrations are low. We have referred to this explanation in the manuscript with the following:

“Therefore, there may be a relationship between 25OHD concentration and iron metabolism when 25OHD concentrations are lowest and not at higher concentrations.”

“However, whilst vitamin D status in early pregnancy and in the supplemented group in late pregnancy were similar to the Thomas study, late pregnancy vitamin D status in the placebo group was notably lower, and further supports a concentration-dependent relationship between vitamin D and iron metabolism”

“Finally, 94% of women in this sub-study were of white, self-reported ethnicity and the participants were screened for low and high 25OHD concentration before inclusion in the trial. It is possible that a larger effect on hepcidin would be detected at lower 25OHD concentrations”.

Minor comment:

Lines 180-182: One assumes that this sentence refers to the P-values in the right column being a comparison of the placebo-group and the VitD3 group – but it is not stated.

Line 282: Another limitation that the authors may wish to include is the non-measurement of 1,25-dihydroxyvitamin D, especially given the mention of this hormonal form of vitamin D in Lines 49-52.

We have made the following changes in light of these comments:

“P-value obtained from 2-sample t-test for normally distributed variables, Mann-Whitney U-tests for non-normally distributed variables and chi-squared tests for categorical variables to test for differences between groups at late pregnancy”

“Secondly, other biomarkers of iron status, erythropoiesis, inflammation or vitamin D (i.e. 1,25(OH)2D) were not measured and so alternative mechanisms were not explored.”
